# An axon-specific expression of HCN channels catalyzes fast action potential signaling in GABAergic interneurons

Fabian C. Roth [ID] [1] & Hua Hu [ID] [1✉]

During high-frequency network activities, fast-spiking, parvalbumin-expressing basket cells (PV⁺-BCs) generate barrages of fast synaptic inhibition to control the probability and precise timing of action potential (AP) initiation in principal neurons. Here we describe a subcellular specialization that contributes to the high speed of synaptic inhibition mediated by PV⁺-BCs. Mapping of hyperpolarization-activated cyclic nucleotide-gated (HCN) channel distribution in rat hippocampal PV⁺-BCs with subcellular patch-clamp methods revealed that functional HCN channels are exclusively expressed in axons and completely absent from somata and dendrites. HCN channels not only enhance AP initiation during sustained high-frequency firing but also speed up the propagation of AP trains in PV⁺-BC axons by dynamically opposing the hyperpolarization produced by $Na^+$-$K^+$ ATPases. Since axonal AP signaling determines the timing of synaptic communication, the axon-specific expression of HCN channels represents a specialization for PV⁺-BCs to operate at high speed.

[1] Division of Physiology, Department of Molecular Medicine, Institute of Basic Medical Sciences, University of Oslo, Oslo, Norway. ✉email: huah@medisin.uio.no

Parvalbumin-expressing basket cells (PV[+]-BCs) represent a key type of GABAergic interneuron in the hippocampus and neocortex, and they make important contributions to a broad range of cortical functions[1,2]. PV[+]-BCs can be functionally identified by the fast-spiking phenotype, in which the interneuron maintains the discharge of brief action potentials (APs) at tens to hundreds of hertz with little spike frequency adaptation[3–5]. These high-frequency APs, which are initiated in the proximal axon[6–8], are translated into the release of GABA from distal presynaptic terminals with remarkably high speed and reliability[9]. During high-frequency network activities, these functional hallmarks provide PV[+]-BCs with the ability to generate barrages of rapid inhibitory outputs to control the probability and precise timing of AP initiation in principal neurons[10,11].

Previous studies have revealed a series of signaling mechanisms that enable PV[+]-BC axons to produce fast and strong inhibition[1,2], including a high density of Na[+] channels to ensure fast AP propagation in the thin and extensively branching interneuron axons[7]. To prevent intracellular Na[+] accumulation during repetitive firing, PV[+]-BCs densely express Na[+]-K[+] ATPases[12,13]. Based on the stoichiometry of the Na[+] pump[14], the combination of the fast-spiking phenotype and the dense expression of Na[+]-K[+] ATPases is expected to generate a strong Na[+] pump-mediated hyperpolarization that can potentially slow down the generation and propagation of APs in PV[+]-BC axons in an activity-dependent manner[15]. In contrast to this prediction, PV[+]-BCs display little spike frequency adaptation even during sustained high-frequency firing[3]. Furthermore, subcellular patch-clamp recordings have revealed that the speed and reliability of AP propagation in PV[+]-BC axons are maintained during a long train of high-frequency APs[7].

To identify the underlying mechanism, we hypothesize that fast AP signaling in PV[+]-BC axons critically depends on an axonal expression of hyperpolarization-activated cyclic nucleotide-gated (HCN) channels[15–17]. In contrast to other types of voltage-gated channels, HCN channels are activated by hyperpolarizations and conduct a non-inactivating depolarizing current[18]. This unique feature makes HCN channels well-suited for opposing the hyperpolarizing Na[+] pump current[19–21]. With subcellular patch-clamp techniques, we show that functional HCN channels in PV[+]-BCs are exclusively expressed in the axonal membrane. There, HCN channels enhance AP initiation during sustained high-frequency firing and facilitate the propagation of APs by dynamically counterbalancing the hyperpolarizing current mediated by Na[+]-K[+] ATPases.

## Results

**Functional HCN channels in PV[+]-BCs are exclusively axonal**. Although previous studies have detected HCN channel expression in PV[+]-BC axons[22–26], the distribution pattern and density of functional HCN channels in this key type of interneuron remained unknown. To address these issues, we mapped the subcellular distribution and quantified the density of HCN channels in hippocampal PV[+]-BCs with the outside-out patch-clamp method (Fig. 1a–d). In membrane patches excised from axons, a 1-s hyperpolarizing voltage pulse evoked an inward current that displayed several hallmark features of h-currents (Fig. 1b). First, the steady-state activation curve indicated that the inward current was activated by membrane potential hyperpolarizations below the AP threshold (Supplementary Fig. 1a, b). Second, the inward current did not inactivate and displayed slow activation and deactivation rates (Supplementary Fig. 1c, d). Third, the current was resistant to barium (all patches recorded with 1 mM barium in the extracellular solution)[27] but sensitive to cesium (4 mM extracellularly applied CsCl; Supplementary

Fig. 1e–g). These results collectively indicate that the inward current is produced by the activity of HCN channels in PV[+]-BC axons. In sharp contrast, the same hyperpolarizing voltage pulse failed to evoke similar inward currents in somatic and dendritic outside-out patches (Fig. 1b). Furthermore, cesium had no effect on currents in somatic and dendritic patches (steady-state somatic current amplitude $= -1.3 \pm 0.4$ pA in control and $-1.5 \pm 0.5$ pA in CsCl, $n = 6$ somatic recordings, $P > 0.99$, two-sided Wilcoxon signed rank test; steady-state dendritic current amplitude $= -2.0 \pm 1.0$ pA in control and $-2.5 \pm 0.8$ pA in CsCl, $n = 6$ dendritic recordings at distances between 10 and 350 μm, $P > 0.99$, two-sided Wilcoxon signed rank test), suggesting that HCN channels are absent from PV[+]-BC somata and dendrites. In close agreement, comparison of axonal h-conductance ($g_h$) density with somatic and dendritic values confirmed that functional HCN channels in PV[+]-BCs were exclusively axonal (Fig. 1c). Within the axon, $g_h$ density was uncorrelated with distance from the soma ($n = 48$ axonal patches, Spearman $\rho = 0.26$, $P > 0.05$, Fig. 1d), indicating a relatively uniform axonal HCN channel distribution.

**HCN channels enhance AP initiation in PV[+]-BCs**. To determine the role of HCN channels in the fast-spiking phenotype, we injected a 1-s depolarizing current pulse at the soma to evoke high-frequency firing. Blocking HCN channels with ZD7288 reduced the steady-state AP frequency to 81% of the control value (Fig. 2a–d, from $192.8 \pm 7.0$ Hz in control to $155.5 \pm 8.7$ Hz in 20 μM ZD7288, $n = 11$ somatic recordings, $P = 0.004$, two-sided Wilcoxon signed rank test). Comparison of input–output relationship in control with that in ZD7288 revealed that ZD7288 impaired high-frequency (>100 Hz) firing without changing the rheobase current required for generating APs (Fig. 2e). These results highlight the contribution of HCN channels to the characteristic fast-spiking phenotype of PV[+]-BCs.

A key determinant of AP initiation probability is the voltage difference between resting membrane potential (RMP) and AP threshold. In agreement with previous results[22], we found that blocking h-currents with ZD7288 hyperpolarized the PV[+]-BC somatic RMP by 2.7 mV (Supplementary Table 1). To test whether ZD7288 impaired high-frequency firing by hyperpolarizing the RMP, we injected constant depolarizing currents at the soma to compensate for the ZD7288-induced RMP hyperpolarization. However, this approach failed to prevent the reduction of AP frequency by ZD7288. With the compensation, ZD7288 reduced the steady-state AP frequency to 79% of the control value (from $202.5 \pm 8.4$ Hz in control to $160.0 \pm 7.5$ Hz in 20 μM ZD7288, $n = 12$ somatic recordings, $P = 0.004$, two-sided Wilcoxon signed rank test), suggesting that HCN channels enhance the high-frequency firing with mechanisms other than exerting a tonic depolarizing influence on the RMP.

Subsequent analyses showed that the impairment of high-frequency firing by ZD7288 was accompanied by an elevation of somatic voltage threshold (Fig. 2f–i). When compared with that in control, the somatic voltage threshold during high-frequency APs progressively increased in ZD7288 (Fig. 2f, h, i). Correlation analyses revealed that the magnitude of the ZD7288-induced AP frequency reduction strongly depended on the elevation of the somatic voltage threshold by ZD7288 (Fig. 2j). These results indicate that HCN channels facilitate the generation of high-frequency APs in PV[+]-BCs by stabilizing the somatic voltage threshold during repetitive firing.

**HCN channels ensure fast and reliable axonal AP propagation**. To determine how HCN channels contribute to AP propagation in PV[+]-BC axons, we obtained simultaneous soma–axon

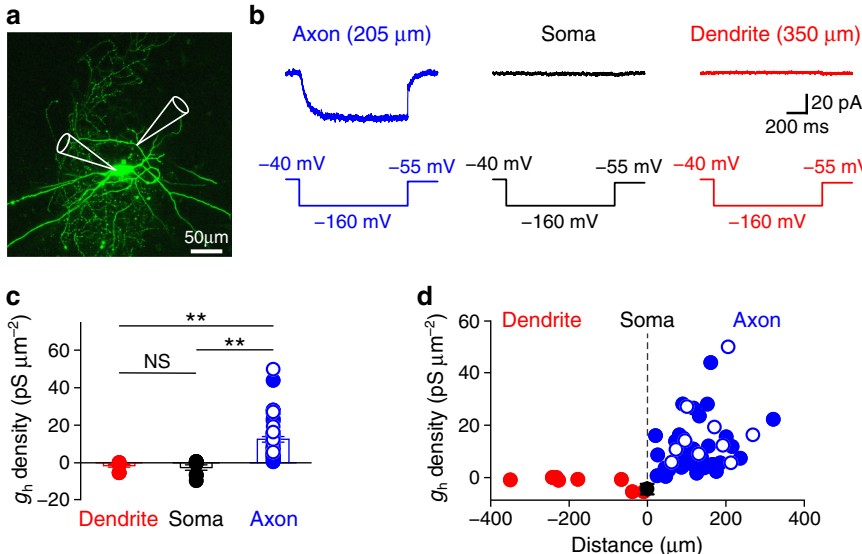

**Fig. 1 HCN channels in PV$^+$-BCs are exclusively expressed in axons. a** Maximal confocal stack projection of a PV$^+$-BC in the dentate gyrus filled with Alexa Fluor 488. Recording pipettes are drawn schematically. Confocal images were acquired for all subcellular recordings ($n = 109$ PV$^+$-BCs). **b** Currents (top) evoked by a hyperpolarizing voltage pulse for activating h-currents (bottom) in outside-out patches excised from either a PV$^+$-BC axon (blue), soma (black), or dendrite (red). **c** Summary graph of h-conductance density in PV$^+$-BC dendrites ($n = 7$ apical and 1 basal dendritic patches), somata ($n = 8$ somatic patches), and axons ($n = 48$ axonal patches). ** indicates $P = 10^{-5}$, NS indicates $P = 0.71$ (two-sided Wilcoxon rank sum test). **d** Summary plot of h-conductance density in **c** against distance from the soma. Open blue circles in **c** and **d**, data obtained from axon shafts; filled blue circles, data obtained from axonal blebs. Error bars in this figure represent ± SEM. Source data are provided as a Source data file.

current-clamp recordings. We evoked APs by injecting a train of short depolarizing current pulses at the soma and simultaneously recorded somatic and axonal voltage responses. To mimic AP patterns under in vivo conditions, we elicited APs with a physiologically relevant spike pattern that was recorded from a hippocampal PV$^+$-BC in a freely moving rat[28]. Analyses of the latency between somatic and corresponding axonal APs revealed that ZD7288 delayed the arrival of APs in distal axons (Fig. 3a–e). Similarly, recordings of unitary inhibitory postsynaptic currents (IPSCs) from pairs of a presynaptic PV$^+$-BC and a postsynaptic granule cell (GC) demonstrated that ZD7288 increased the latency between presynaptic APs and IPSCs (Supplementary Fig. 2). To quantify axonal AP propagation speed, we plotted the latency between somatic and axonal APs against the distance between recording sites (Fig. 3f). Fitting the latency–distance relationship with a bilinear function revealed that ZD7288 reduced the AP propagation speed in PV$^+$-BC axons (Fig. 3g). ZD7288 also produced a small shift of AP initiation site towards the soma (Fig. 3h), but this shift is too small to account for the delay of AP arrival in distal axons caused by ZD7288. To test whether the effect of ZD7288 on AP propagation depends on specific features of the physiologically relevant AP pattern, we subsequently analyzed the propagation of a train of 300 APs with a constant frequency of 20 Hz (Supplementary Fig. 3). These recordings gave qualitatively similar results, providing additional evidence to support the idea that HCN channels are important for the fast propagation of AP trains in PV$^+$-BC axons.

After blocking HCN channels with ZD7288, we frequently detected spikelets in distal axons (Fig. 4a). Superimposition of these spikelets and the corresponding somatic voltage revealed that each spikelet was consistently associated with a full-blown somatic AP (Fig. 4b, c), indicating that the spikelets were a direct consequence of axonal AP propagation failures. By contrast, propagation failures were not observed under control conditions (Fig. 4d). Based on these results, we conclude that HCN channels increase both the speed and reliability of AP propagation in PV$^+$-BC axons.

**Axonal HCN channels oppose hyperpolarizing Na$^+$-pump currents.** We next sought the mechanism by which HCN channels facilitate the initiation and propagation of AP trains. While a recent paper shows that HCN channels accelerate AP propagation in cerebellar mossy fibers by tonically depolarizing the RMP[17], other studies suggest that HCN channels increase axonal excitability during repetitive firing by dynamically opposing hyperpolarizing Na$^+$-K$^+$ ATPase currents induced by APs[19–21]. To test whether these mechanisms are shared by PV$^+$-BC axons, we analyzed the effect of ZD7288 on subthreshold axonal membrane potential dynamics. We found that ZD7288 hyperpolarized the axonal RMP by 2.9 mV and increased the axonal input resistance (Supplementary Table 1). Furthermore, comparison of axonal length constant in control with that in ZD7288 revealed that blocking HCN channels enhanced subthreshold voltage transfer from the soma to the axon (Supplementary Table 1 and Supplementary Fig. 4). In contrast to its relatively small effect on the axonal RMP, ZD7288 unmasked a pronounced axonal hyperpolarization during repetitive firing (Fig. 5a–c and Supplementary Fig. 5), which built up slowly during the AP train and evolved into a long-lasting (>5 s) afterhyperpolarization after the train. Detailed analyses of axonal membrane potential trajectories during repetitive firing revealed that PV$^+$-BC axons reached a maximum negative potential of $-88.8 \pm 2.3$ mV (ranging between $-77.7$ and $-101.7$ mV in 12 soma–axon recordings) after inhibiting HCN channels with 20 μM ZD7288 (Fig. 5a). The ZD7288-induced hyperpolarization disappeared after blocking AP generation with 1 μM tetrodotoxin (TTX; peak amplitude = $12.15 \pm 3.52$ mV in 20 μM ZD7288 and $-0.09 \pm 0.28$ mV during the co-application of 20 μM ZD7288 and 1 μM TTX, 3 soma–axon recordings), indicating its AP dependence. Comparing the amplitude of the axonal AP-dependent hyperpolarization with that at the soma revealed an axonal origin (Fig. 5d, e). We found that the amplitude of the AP-dependent hyperpolarization was independent of axonal membrane potential, indicating that it is not mediated by K$^+$ channels (Supplementary Fig. 5d, e). However, application of the Na$^+$-K$^+$ ATPase blocker ouabain fully

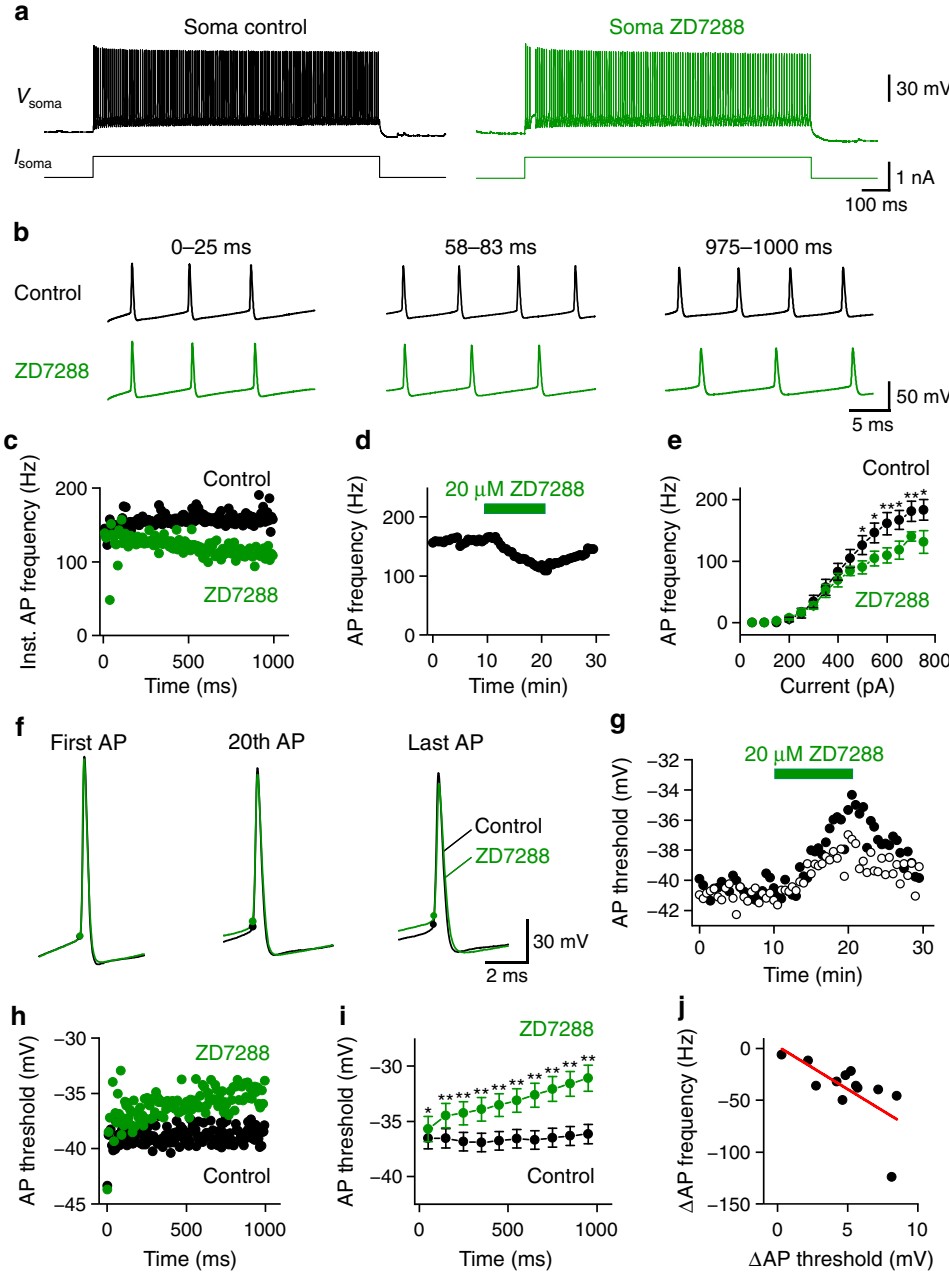

**Fig. 2 HCN channels enhance AP initiation during sustained high-frequency firing. a** Somatic recording of AP trains evoked by injecting a 1-s current pulse into the soma of a PV+-BC in control (black) and 20 μM ZD7288 (green). **b** Membrane potentials in **a** plotted on an expanded time scale to compare AP frequencies in three different time windows (0–25, 58–83, and 975–1000 ms from the onset of the current pulse). **c** Instantaneous AP frequency in control (black) and ZD7288 (green) plotted against the time from the onset of the current pulse to show spike frequency adaptation in ZD7288. **d** Steady-state AP frequency plotted against experimental time. **e** Summary plot comparing the input–output relationship of PV+-BCs in control (black) with that in 20 μM ZD7288 (green). $n = 12$, 11, 10, and 9 PV+-BCs, when the somatic depolarizing current pulse amplitude was 50–550, 600, 650–700, and 750 pA, respectively. **f** First, the 20th and last APs from the spike train in **a** in control (black) superimposed over those in ZD7288 (green). Filled circles indicate respective AP threshold. **g** Somatic voltage threshold of the 20th (open circles) and last (filled circles) APs in the train plotted against experimental time. **h** Somatic voltage threshold from the spike trains in **a** plotted against time from the onset of the 1-s current pulse. Data in **a–d** and **f–h** are from the same recording. **i** Same as in **h**, but shows the summary from 13 PV+-BCs in control (black) and 20 μM ZD7288 (green). For clarity, we divided the 1-s train into 10 equally spaced time bins and calculated the mean threshold value of each bin ($n = 13$ PV+-BCs). **j** Relationship between the 20 μM ZD7288-induced AP frequency reduction and the elevation of somatic voltage threshold. Data were fit with a linear function (red line, Spearman $\rho = -0.73$, $P = 0.01$, $n = 12$ PV+-BCs). Error bars in this figure represent ± SEM. * indicates $P = 0.01$–0.04, ** indicates $P = 0.002$–0.009 in two-sided Wilcoxon signed rank test. Source data are provided as a Source data file.

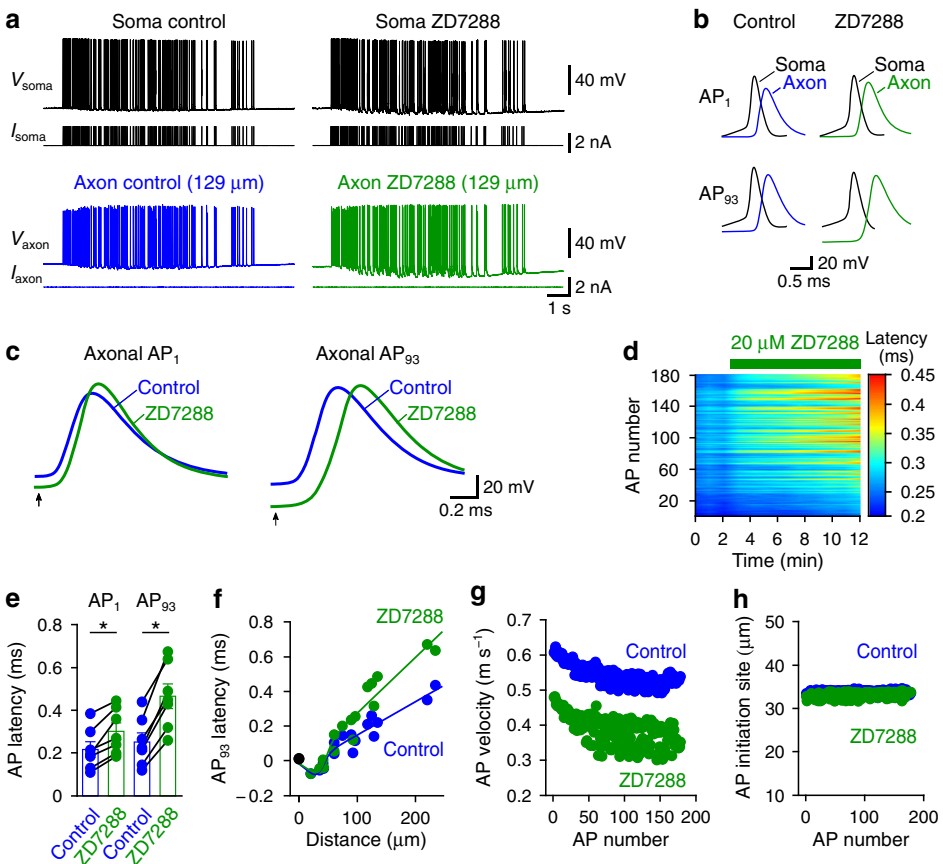

**Fig. 3 HCN channels speed up the propagation of a physiologically relevant AP pattern in PV$^+$-BC axons. a** Simultaneous soma–axon recording of APs elicited by stimulating the soma of a PV$^+$-BC with a physiologically relevant AP pattern in control (left) and 20 µM ZD7288 (right). **b** First (top, AP$_1$) and 93rd (bottom, AP$_{93}$) APs in the train in control and ZD7288 plotted on an expanded time scale. **c** Superimposition of axonal APs in control and ZD7288 to show that blocking HCN channels delayed the arrival of APs at the axonal recording site. Only the first and 93rd APs in the train are shown. To compare spike timing, axonal APs were aligned by using the time point (arrows) at which the corresponding somatic AP reached the half-maximal peak amplitude in the rising phase as the temporal reference. Black traces in **a**–**c**, somatic voltage and current; blue traces, axonal voltage and current in control; green traces, axonal voltage and current in ZD7288. **d** Heat map plotting the latency between somatic and corresponding axonal APs for every AP in the train against experimental time. Data in **a**–**d** are from the same experiment. **e** Summary graph showing that 20 µM ZD7288 increased the soma–axon AP latency in 7 distal (>100 µm) recordings. For clarity, only the first and 93rd AP latencies are shown. Data points from the same recording are connected by lines. * indicates $P = 0.02$ ($n = 7$ soma–axon recordings, two-sided Wilcoxon signed rank test). Error bars in bar graphs represent ± SEM. **f** Latency between the 93rd somatic AP in the train and corresponding axonal AP plotted against distance from the soma ($n = 16$ soma–axon recordings). Data were fit with a bilinear function (continuous lines). **g**, **h** Axonal propagation velocity (**g**) and initiation site (**h**) of every AP in the train in control and 20 µM ZD7288. Blue circles and lines in **e**–**h**, from data in control; green circles and lines, from data in 20 µM ZD7288. Source data are provided as a Source data file.

inhibited the AP-dependent hyperpolarization (Fig. 5f, g). These results support the idea that HCN channels enhance PV$^+$-BC axonal excitability during repetitive firing mainly by dynamically opposing the hyperpolarizing Na$^+$ pump current.

To determine the direct effect of hyperpolarization on the speed and reliability of AP propagation in PV$^+$-BC axons, we injected a constant hyperpolarizing current into the axon without blocking HCN channels. We found that axonal hyperpolarization evoked by the current injection alone increased the latency between somatic and axonal APs (Supplementary Fig. 6), indicating that axonal hyperpolarizations are sufficient to slow down AP propagation. However, this approach did not induce AP propagation failures over a broad range of axonal membrane potentials (from −68 mV to −92 mV, 7 distal soma–axon recordings), in stark contrast to the frequent occurrence of AP propagation failures after blocking HCN channels with ZD7288. A likely explanation is that the short length constant of PV$^+$-BC axons (Supplementary Table 1 and Supplementary Fig. 4) prevents the hyperpolarization induced by the negative current

injection from spreading to axonal structures that have a low safety factor for successful AP propagation[15]. By contrast, blocking HCN channels with ZD7288 is expected to produce a more global axonal hyperpolarization because of the relatively uniform axonal HCN channel distribution.

## Discussion

The precise subcellular distribution pattern of ion channels is an important determinant of their functions[29]. Several types of principal neurons express a high density of HCN channels in their dendrites where these channels control dendritic electrogenesis, synaptic integration and synaptic plasticity[27,30–34]. In contrast to the wealth of information gained from principal cells, the distribution and function of HCN channels in GABAergic interneurons at the subcellular level are less well understood. Taking advantage of subcellular patch-clamp methods, this study provides a detailed description of HCN channel subcellular distribution pattern in a major type of cortical interneuron. Our

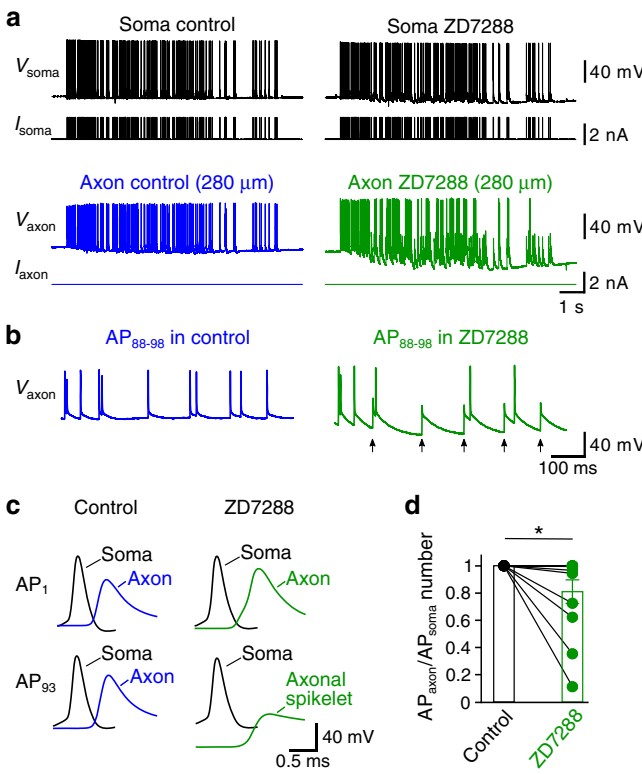

**Fig. 4 HCN channels ensure reliable propagation of a physiologically relevant AP pattern in PV$^+$-BC axons. a** Simultaneous soma–axon recording of APs elicited by stimulating the soma with a physiologically relevant spike pattern in control (left) and 20 μM ZD7288 (right). The distance between axonal and somatic sites was 280 μm. **b** Axonal membrane potentials in **a** between the 88th and 98th somatic APs (AP$_{88-98}$) plotted on an expanded time scale. Black arrows indicate axonal propagation failures in ZD7288. **c** First (top, AP$_1$) and 93rd (bottom, AP$_{93}$) APs of the AP trains in **a** plotted on an expanded time scale. Note that the 93rd somatic AP in ZD7288 is associated with a spikelet at the axonal recording site, indicating an AP propagation failure. Black traces in **a**–**c**, somatic voltage and current; blue traces, axonal voltage and current in control; green traces, axonal voltage and current in ZD7288. **d** Summary graph showing that 20 μM ZD7288 reduced the propagation reliability of the physiologically relevant AP pattern. AP propagation reliability was quantified by dividing the number of full-blown APs in the axon by that in the soma during the spike train. A value of less than one indicates AP propagation failures. AP propagation failures were detected in 7 out of 12 distal (>100 μm) soma–axon recordings after applying ZD7288. Data points from the same experiment are connected by lines. * indicates $P = 0.02$ ($n = 12$ soma–axon recordings, two-sided Wilcoxon signed rank test). Error bars represent ± SEM. Source data are provided as a Source data file.

results have revealed that functional HCN channels in PV$^+$-BCs are exclusively expressed in axons and are absent from the somatodendritic membrane. This axon-specific subcellular distribution pattern sets PV$^+$-BCs apart from many other types of central neurons, ranging from neocortical layer 5 pyramidal neurons to cerebellar Purkinje neurons, in which HCN channels have been detected at the soma and dendrites[27,31,33,35–38].

Our results show that HCN channels in PV$^+$-BC axons not only enhance AP initiation during sustained firing but also facilitate the propagation of APs. Previous studies have demonstrated that HCN channels modulate neuronal excitability in complex ways[39]. Activation of HCN channels creates a depolarizing influence that

brings the RMP closer to the AP threshold, but the same HCN channels also generate a shunting conductance that reduces the input resistance and neuronal excitability[40]. Furthermore, the depolarization produced by HCN channels modulates functional states of Na$^+$, K$^+$, and Ca$^{2+}$ channels that are critical for AP signaling[37,41–44]. These multiple actions of HCN channels make it challenging to predict the net effect of HCN channels on neuronal excitability in a cell type-specific manner. Indeed, experimental evidence from different types of axons has confirmed that HCN channels can either enhance or inhibit axonal AP signaling[17,20,37,45]. Here, our subcellular recordings provide direct evidence to indicate that HCN channels in PV$^+$-BC axons exert a net excitatory effect in terms of AP initiation and propagation. It appears that HCN channels increase PV$^+$-BC axonal excitability mainly by dynamically opposing the activity-dependent hyperpolarization created by Na$^+$-K$^+$ ATPases during repetitive firing and, to a lesser degree, by tonically depolarizing the RMP. Collectively, these two mechanisms facilitate AP initiation and propagation by keeping the axonal membrane potential close to the Na$^+$ channel activation threshold. Blocking HCN channels raises the somatic voltage threshold for initiating APs because the soma needs to supply the AP initiation site in the proximal axon with more charges to overcome the hyperpolarization in ZD7288. Likewise, ZD7288 impairs AP propagation because it is more difficult for the distal axon to reach the Na$^+$ channel activation threshold from more negative potentials.

What is the functional advantage of expressing HCN channels exclusively in axons? Interestingly, the axon-specific expression of HCN channels in PV$^+$-BCs is matched by a similarly polarized subcellular distribution of voltage-gated Na$^+$ channels[1,7]. Consequently, 94% of AP-induced Na$^+$ entry in PV$^+$-BCs occurs in the axon[46]. To maintain ionic homeostasis during repetitive firing, PV$^+$-BC axons densely express Na$^+$-K$^+$ ATPases. Consistent with immunolabeling data[12], our subcellular recordings indicate an axonal origin of the Na$^+$ pump-mediated hyperpolarization. Our data also revealed a relatively short length constant of PV$^+$-BC axons, which restricts the interaction between spatially segregated HCN channels and Na$^+$-K$^+$ ATPases. Axonal expression of HCN channels thus increases the sensitivity to detect and the efficiency to counteract the hyperpolarizing Na$^+$ pump current during repetitive firing.

HCN channels in PV$^+$-BC axons may make important contributions to cortical network functions. For controlling network dynamics, both the magnitude and timing of synaptic inhibition are important[47]. Our results identified HCN channels as an important component of the subcellular signaling machinery that enables PV$^+$-BCs to generate strong and fast inhibition[1,2]. Our recordings indicate that HCN channels contribute to the robust initiation and reliable propagation of APs in PV$^+$-BC axons. Inhibiting HCN channels in PV$^+$-BCs thus may tip the excitation-inhibition balance inside the network by reducing the amount of inhibition produced by this key type of interneuron. In addition, we found that HCN channels control the timing of inhibition produced by PV$^+$-BCs by speeding up axonal AP propagation. Based on our analyses and PV$^+$-BC axonal morphology[48], we estimate that blocking HCN channels can increase the latency between AP initiation in the proximal axon and release of GABA from the most distal axon terminals by over 1 ms during repetitive firing. Short-latency inhibitory outputs from PV$^+$-BCs narrow the time window of temporal summation and expand the dynamic range of pyramidal neurons, as well as contribute to network oscillations in the gamma frequency range[49–51]. By accelerating AP propagation, HCN channels in PV$^+$-BC axons may play a key role in implementing these network functions. Finally, PV$^+$-BCs co-express HCN1 and

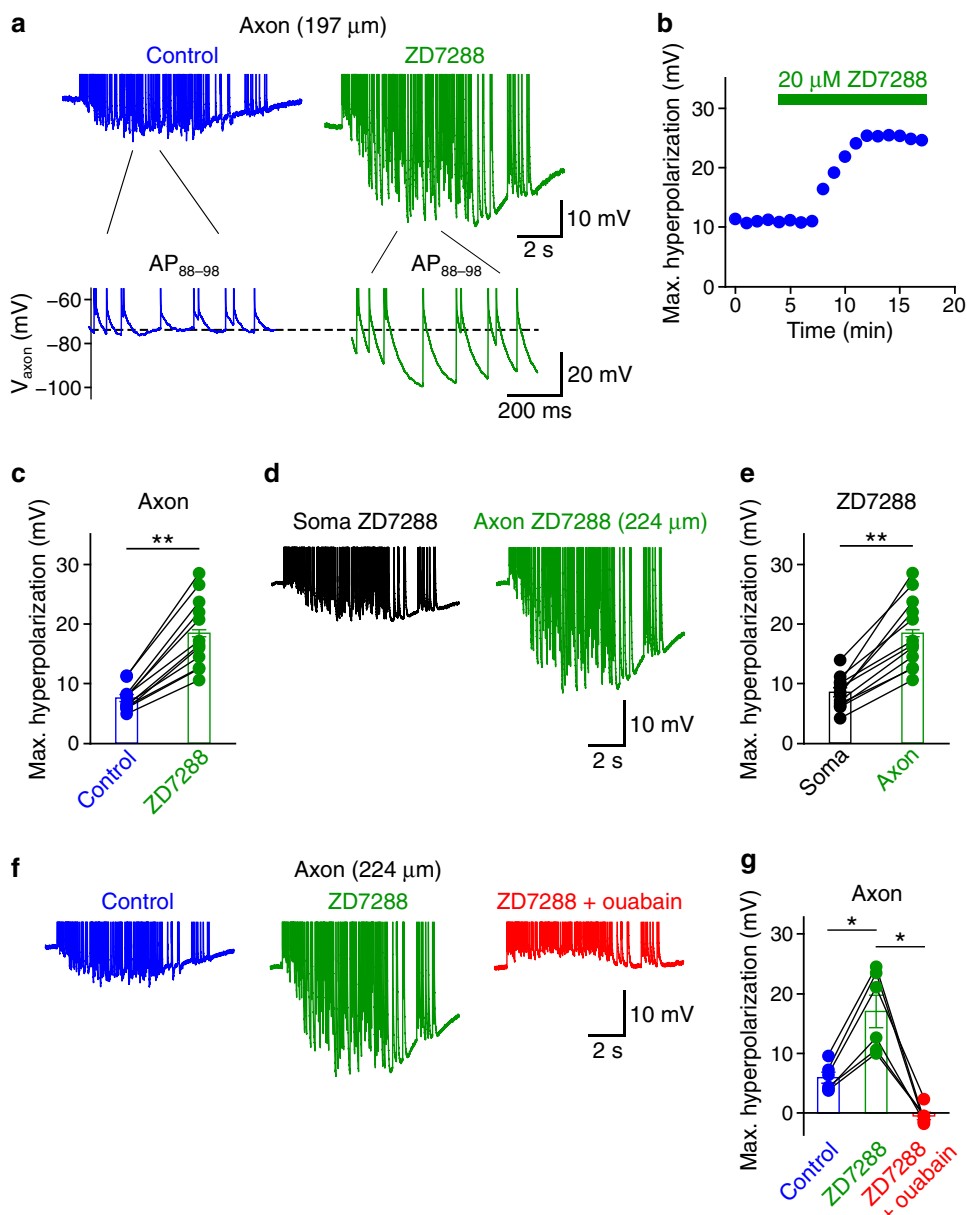

**Fig. 5 HCN channels oppose the hyperpolarization created by Na$^+$-K$^+$ ATPases in PV$^+$-BC axons. a** Top, subthreshold axonal voltage trajectories during the physiologically relevant AP train in control (blue) and 20 μM ZD7288 (green). Bottom, axon voltage trajectories between the 88th and 98th AP (AP$_{88-98}$) plotted on an expanded time scale. APs in this figure were truncated at −55 mV to highlight hyperpolarizations between APs in ZD7288. To facilitate the comparison with other recordings, we adjusted the baseline axonal membrane potential in control to −66 mV by injecting a constant hyperpolarizing current of −19.6 pA into the axon during the soma–axon recording. This holding current was not changed throughout the experiment. **b** Peak amplitude of the axonal AP-dependent hyperpolarization (Max. hyperpolarization) from the recording in **a** plotted against experimental time. **c** Summary graph showing the peak amplitude of the AP-dependent hyperpolarization in PV$^+$-BC axons in control and 20 μM ZD7288. ** indicates $P = 0.0005$ ($n = 12$ soma-axon recordings, two-sided Wilcoxon signed rank test). **d** Subthreshold somatic voltage trajectory and corresponding axonal voltage signal in a PV$^+$-BC in 20 μM ZD7288. **e** Summary graph comparing the peak amplitude of the AP-dependent hyperpolarization in PV$^+$-BC somata in 20 μM ZD7288 with that of the corresponding axonal signal. ** indicates $P = 0.0005$ ($n = 12$ soma-axon recordings, two-sided Wilcoxon signed rank test). **f** Subthreshold axonal voltage trajectory during the physiologically relevant AP pattern in control (blue), 20 μM ZD7288 (green) and after co-application of 20 μM ZD7288 and 200 μM ouabain (red) from a soma–axon recording. **g** Summary graph showing the peak amplitude of the AP-dependent hyperpolarization in six PV$^+$-BC axons in control, 20 μM ZD7288 and after co-application of 20 μM ZD7288 and 200 μM ouabain. * indicates $P = 0.03$ ($n = 6$ soma-axon recordings, two-sided Wilcoxon signed rank test). Axonal voltage traces in **d** and **f** are from the same recording. In **c**, **e**, and **g**, data points from the same experiment are connected by lines. Error bars in this figure represent ± SEM. Source data are provided as a Source data file.

HCN2 subunits[22]. With the high sensitivity of HCN2 subunits to cAMP[52], neuromodulation of HCN channels in PV$^+$-BC axons may represent a mechanism by which the brain fine-tunes the magnitude and timing of synaptic inhibition[17].

## Methods

**Patch-clamp recordings from PV$^+$-BC axons and dendrites**. Experiments on Wistar rats were ethically approved by the Norwegian Food Safety Authority (Mattilsynet) and were performed in strict accordance with institutional, national, and European guidelines for animal experimentation.

Subcellular recordings from fast-spiking PV+-BCs of the dentate gyrus were performed with the following experimental protocol. Transverse hippocampal slices (thickness 350 μm) were prepared from brains of 17- to 23-day-old male Wistar rats. Rats were housed under a 12 h light (7 a.m.–7 p.m.) and dark (7 p.m.–7 a.m.) cycle and were kept in a litter of eight to ten animals together with the mother in a single cage. Slices were cut in ice-cold, sucrose-containing physiological extracellular solution using a vibratome (VT1200, Leica Microsystems), incubated in a storage chamber filled with standard physiological extracellular solution at ~34 °C for 30 min and subsequently stored at room temperature. Slices were then individually transferred into a recording chamber and superfused with standard physiological extracellular solution. Current-clamp recordings were performed at near-physiological temperature (~33 °C; range: 31 °C–34 °C). The bath solution was heated with a resistive heating unit (Sigmann Elektronik, Hüffenhardt, Germany) prior to entering the recording chamber, and the recording temperature was continuously monitored with a micro thermistor positioned in the vicinity of the slice. For biophysical analyses of h-conductance gating and distribution (Fig. 1 and Supplementary Fig. 1), recordings were made at room temperature (~24 °C; range: 22 °C–25 °C) to maximize the precision of kinetic measurements. In each experiment, the temperature was held constant within ±0.5 °C.

For recordings from interneuron axons and dendrites, we used the following experimental strategy. First, a somatic recording was obtained using an internal solution containing Alexa Fluor 488 (50 or 100 μM, Invitrogen). Second, after ~30 min of somatic whole-cell recording, the fluorescently labeled axon and dendrites were traced from the PV+-BC soma with a Nipkow spinning disk confocal microscope (Volocity, Perkin Elmer, equipped with an Orca camera, Hamamatsu, and a solid-state laser, excitation wavelength 488 nm). Total exposure time was minimized to avoid phototoxic damage. Finally, fluorescent and infrared differential interference contrast (IR-DIC) images were compared and axons or dendrites were patched under IR-DIC. Axonal recordings were made from either axon shafts or small spherical axon expansions, presumably representing blebs formed during the slicing procedure[53], at distances up to 331 μm from the soma. Axonal whole-cell recordings were readily obtained, consistent with minimal myelination of PV+-BC axons in the dentate gyrus at the developmental stage used[7,54,55]. This procedure resulted in a simultaneous soma–axon recording configuration. In all cases, axons could be unequivocally distinguished from dendrites on the basis of smaller diameter, location within or adjacent to the granule cell layer, and abundance of tangential collaterals. Outside-out patches were obtained by slowly withdrawing the patch pipette after establishing the axonal or dendritic whole-cell recording configuration. In agreement with previous results[7], conductance density values were similar between outside-out patches excised from axonal blebs and those excised from shafts (Fig. 1c, d). In recordings to determine the somatic AP voltage threshold (Fig. 2), current injection and voltage measurement were performed separately with two somatic patch electrodes to avoid artifacts from imperfect series resistance compensation.

Patch pipettes were fabricated from thick-walled borosilicate glass capillaries (outer diameter: 2 mm, inner diameter: 1 mm) with a horizontal pipette puller (P-97, Sutter Instruments). When filled with internal solution, they had a resistance of 2–10 MΩ for somatic recordings and 6–40 MΩ for axonal and dendritic recordings. Current- and voltage-clamp recordings were performed using a Multiclamp 700B amplifier (Molecular Devices). Series resistance in current-clamp recordings was 12–90 MΩ. Cells with somatic resting potentials more positive than −50 mV were discarded. Pipette capacitance and series resistance compensation (bridge balance) were applied throughout current-clamp experiments. Pipette capacitance compensation in current-clamp recordings was performed by adjusting the pipette capacitance neutralization function of the amplifier to the highest value without provoking oscillations inside the amplifier circuit, and bridge balance was monitored and readjusted as required during the course of each experiment. PV+-BCs under control conditions were adjusted to a membrane potential of approximately −65 mV by injecting a holding current at the soma (range: −150 to +200 pA) or at the axon (range: −100 to −10 pA). The holding currents were not changed throughout the experiment except where indicated.

Signals were low-pass filtered at 10 kHz in current-clamp recordings and at 2 or 4 kHz in voltage-clamp recordings and sampled at 50 or 100 kHz with a Digidata 1322 converter board (Molecular Devices). Pulse protocols were generated using pClamp 9 or 10 (Molecular Devices). Voltage protocols were applied to outside-out patches once every 20–40 s. Leak and capacitive currents were subtracted online using a P over −8 correction procedure. The holding potential before and after the pulse sequence in voltage-clamp recordings of h-currents was −40 mV.

To analyze axonal AP propagation (Figs. 3–5 and Supplementary Figs. 2, 3, and 5), APs were evoked by injecting a train of brief current pulses into the soma once every 60 s during simultaneous soma–axon or paired PV+-BC–GC recordings. As ZD7288 raised the somatic AP threshold, we adjusted the amplitude of the current pulses to ensure that each pulse faithfully elicited a single AP during the course of the experiment. For illustrative purposes, residual pipette capacitance artifacts at the beginning and end of the voltage response to the injection of the brief current pulse were manually removed from somatic voltage plots in Figs. 3–5 and Supplementary Figs. 2 and 3. The physiologically relevant AP pattern recorded from a hippocampal PV+-BC in a freely moving rat was kindly provided by Dr. Klausberger (cell ID: TV08k in ref. [28]). The spike pattern, which contains 178 APs, encompasses a broad frequency range (1.4–512.8 Hz, mean frequency 21.7 Hz).

PV+-BCs were identified based on the non-accommodating, fast-spiking AP phenotype (steady-state AP frequency >50 Hz at room temperature and >150 Hz at physiological temperature in response to 1-s, 0.3 to 1-nA somatic current pulses), and the morphological properties of the axonal arbor, which was largely restricted to the granule cell layer and established basket-like structures around GC somata in confocal images. In a previous sample of fast-spiking interneurons in the dentate gyrus analyzed in detail by light microscopy, the fast-spiking AP phenotype was tightly correlated with the expression of parvalbumin[56]. Furthermore, 78 of 83 cells were classical basket cells with tangential axon collaterals and basket-like branches around GC somata, whereas 5 out of 83 were axo-axonic cells with radial axon collaterals[56]. Based on these results, the recorded cells were termed PV+-BCs throughout the present study.

**Paired PV+-BC–GC recordings.** To examine the effect of HCN channel block on synaptic latency, paired recordings were made between monosynaptically connected PV+-BCs and GCs (Supplementary Fig. 2). In these experiments, the distance between somata of pre- and postsynaptic neurons was 252.2 ± 29.6 μm and the recording temperature was 31–34 °C. Series resistance in the postsynaptic GC was 7–11 MΩ and remained stable in each experiment. GCs were voltage-clamped at −80 mV without series resistance compensation. Latency was measured from the time point of the half-maximal amplitude in the rising phase of the presynaptic AP to the onset of the corresponding IPSC.

**Solutions and chemicals.** The standard physiological extracellular solution contained 125 mM NaCl, 25 mM NaHCO₃, 2.5 mM KCl, 1.25 mM NaH₂PO₄, 2 mM CaCl₂, 1 mM MgCl₂, and 25 mM D-glucose (equilibrated with 95% O₂ and 5% CO₂ gas mixture). To isolate h-currents in outside-out patch recordings, the standard extracellular solution was replaced by a K+-rich solution containing 120 mM KCl, 10 mM HEPES, 2 mM CaCl₂, 1 mM MgCl₂, 20 mM tetraethylammonium chloride (TEA), 5 mM 4-aminopyridine (4-AP), 0.5 μM TTX, and 1 mM BaCl₂, pH adjusted to 7.4 with KOH[27]. Negative conductance values in a subset of proximal and somatic recordings (Fig. 1c, d) may reflect a contribution from residual K+ currents that were resistant to TEA, 4-AP, and BaCl₂ under our recording conditions. As CsCl had no effect on dendritic and somatic currents, the possibility that the residual current masks h-currents in dendritic and somatic patches can be excluded. Furthermore, the Cs-sensitive conductance density is not significantly different from the $g_h$ density in Fig. 1c and d (Cs-sensitive conductance density = 0.8 ± 0.8 pS μm⁻² in dendrites, $n = 6$ dendritic patches; 0.3 ± 0.7 pS μm⁻² in somata, $n = 6$ somatic patches; 14.5 ± 3.7 pS μm⁻² in axons, $n = 7$ axonal patches, $P = 0.14$–0.66, when comparing the Cs-sensitive conductance density with the $g_h$ density, two-sided Wilcoxon rank sum test), suggesting that the inaccuracy introduced by the residual current to the measurement of $g_h$ density has been minimized under our recording conditions. Whole-axon h-current recordings (Supplementary Fig. 1b) were performed in the standard physiological extracellular solution containing 1 μM TTX, and whole-axon h-currents were isolated by subtracting currents recorded in the presence of 20 μM ZD7288 from those recorded under control conditions. To avoid the off-target effect of CsCl on inward-rectifier K+ channels, we blocked HCN channels with ZD7288 in current-clamp recordings and paired PV+-BC–GC recordings. In a subset of current-clamp recordings, excitatory synaptic transmission was blocked with DNQX and APV. ZD7288 (10–20 μM), CsCl (4 mM), TTX (0.5–1 μM), DNQX (10 μM), and APV (50 μM) were added to the extracellular solution. To avoid the strong depolarization of the RMP and the AP suppression associated with prolonged ouabain application[19,57], we bath-applied ouabain (200 μM) for 30–45 s[13]. Given that the effect of ouabain in acute brain slices is nearly irreversible[19], we compared axonal voltage signals before and immediately after the brief exposure to ouabain to determine the contribution from Na+-K+ ATPases to the AP-dependent hyperpolarization unmasked by ZD7288 (Fig. 5).

The intracellular solution for outside-out patches, dual somatic and soma–axon whole-cell recordings contained 120 mM K-gluconate, 20 mM KCl, 10 mM EGTA, 2 mM MgCl₂, 2 mM Na₂ATP, and 10 mM HEPES, pH adjusted to 7.3 with KOH. Fifty or 100 μM Alexa Fluor 488 was added to the internal solution for all somatic recording electrodes. In paired PV+-BC–GC recordings, the internal solution for the presynaptic PV+-BC contained 135 mM K-gluconate, 20 mM KCl, 0.1 mM EGTA, 2 mM MgCl₂, 2 mM Na₂ATP, and 10 mM HEPES, whereas the solution for the postsynaptic GC contained 145 mM KCl, 0.1 mM EGTA, 2 mM MgCl₂, 2 mM Na₂ATP, and 10 mM HEPES, pH adjusted to 7.3 with KOH.

**Data analysis.** Analyses were performed using Stimfit 0.9.2–0.13.2[58], Clampfit 9 and 10 (Molecular Devices), Origin 2015 (Microcal), Excel 2016 (Microsoft), Minitab 17 (Minitab), and Matlab R2018a (MathWorks). To determine the somatic input–output relationship of PV+-BCs (Fig. 2), we injected a series of 1-s depolarizing current pulses with increasing amplitude (50–750 pA) at the soma and plotted the steady-state frequency of the elicited AP train against the amplitude of the corresponding current pulse. Steady-state AP frequency of the 1-s spike train was computed from the number of APs during the last 100 ms of the train. The AP voltage threshold was defined as the membrane potential at the first data point in the AP depolarizing phase where the rate of depolarization exceeded 50 V s⁻¹ (ref. [59]). The latency between somatic and axonal APs was quantified using the time points corresponding to half-maximal amplitude in the AP rising phase.

In the recordings in which ZD7288 induced axonal AP propagation failures, voltage traces with AP propagation failures were excluded from the analysis of soma–axon AP latency. To determine the AP propagation speed (Fig. 3), latency–distance data were fit with a bilinear function with sigmoidal transition of the form $L(x) = (1 − f(x))ax + f(x)(bx + c)$, where $L$ is latency, $x$ is distance, $a$ and $b$ are slope factors, $c$ is an offset, and $f(x)$ is a sigmoidal (Boltzmann-like) function. AP propagation velocity was obtained as $1/b$ (ref. [7]). The peak amplitude of the AP-dependent hyperpolarization in $PV^+$-BC axons (Fig. 5 and Supplementary Fig. 5) was measured relative to the baseline membrane potential immediately before the AP train. To quantitatively describe steady-state voltage attenuation in $PV^+$-BC axons, we injected a 1-s negative current pulse into either the soma or the axon during simultaneous soma–axon recordings and recorded corresponding voltage responses at both sites (Supplementary Fig. 4). Steady-state voltage transfer was quantified by dividing the steady-state amplitude of the propagated voltage response by that of the response at the site of current injection. Axonal length constant (Supplementary Table 1) was determined by fitting the voltage transfer–distance relationship with a monoexponential function. To minimize any contamination by the delayed, unspecific effect of ZD7288[60], analyses were limited to data acquired within less than 15 min after applying ZD7288.

To facilitate the quantification of h-current amplitudes, we assigned inward currents in voltage-clamp recordings positive amplitude values. Conversely, negative amplitude values were assigned to outward currents. To determine the h-current reversal potential in outside-out patch recordings, we measured the peak amplitude of the tail current evoked by a test voltage pulse to various potentials after a 1-s conditioning pulse to −160 mV. In close agreement with a previous study[27], the tail current reversed at −5.2 ± 1.5 mV in the $K^+$-rich extracellular solution (six axonal outside-out patch recordings). As whole-axon h-currents (Supplementary Fig. 1b) were recorded in the standard physiological extracellular solution (containing 1 μM TTX), the reversal potential of whole-axon h-currents was assumed to be −27.4 mV[22]. To determine h-conductance values in excised patches and whole-axon recordings, we divided the steady-state h-current amplitude by the voltage difference between the test voltage pulse and the h-current reversal potential. H-channel activation curves were determined by calculating the ratio of h-conductance values at various test potentials to the maximum conductance value at −150 mV in each experiment. The activation curve was fit with a Boltzmann function $f = 1/(1 + Exp[(V_{1/2} − V)/k])$, where $V$ is the membrane potential, $V_{1/2}$ is the midpoint potential, and $k$ is the slope factor. H-current activation time constants were obtained by fitting the rising phase of h-currents with a function of the following form $I(t) = a × (1 − Exp[−(t − δ)/τ])$ for $t > δ$ and $I(t) = 0$ for $t ≤ δ$, where $I$ is the current, $t$ is time, $δ$ is a delay, $τ$ is the activation time constant, and $a$ is the amplitude. H-current deactivation time constants were determined by fitting the decay phase of the tail currents with a monoexponential function.

**Quantification and statistical analysis.** Membrane potentials are specified without correction for liquid junction potentials. Values and summary graphs indicate mean ± SEM. Significance of difference between two groups of data was assessed by a two-sided nonparametric Wilcoxon signed rank test or Wilcoxon rank sum test. Statistical difference between axonal length constants was assessed by plotting the voltage transfer on a logarithmic scale against the distance to compare the slope of respective linear regressions in control and ZD7288 (analysis of covariance). Differences were considered significant if $P < 0.05$. Correlation between two variables was assessed with the nonparametric Spearman rank correlation test. Distances in axonal recordings were measured from the point of origin of the axon to the axonal recording site along the axonal trajectory in the confocal maximal intensity stack projection, in which sections at different focal planes had been merged following each experiment. In eight recordings, the axon could not be traced back from the recording site to the soma. These recordings were excluded from the analyses of distance-dependence. Experiments in which the axon originated from one of the dendrites were excluded from the analyses to determine AP propagation speed (five recordings from the analyses in Fig. 3f–h) and axonal voltage transfer (four recordings from the analyses in Supplementary Fig. 4 and Supplementary Table 1). Distances in dendritic recordings were defined as the distance from the dendritic recording site to the center of the soma. The membrane area of outside-out patches was quantified with a previously established relation between patch area and pipette conductance: $A = 0.08271 × g_P − 0.47526$, where $A$ is patch area ($μm^2$) and $g_P$ is pipette conductance (nS)[7]. Conductance density values (Fig. 1c, d) were determined from the steady-state h-current amplitude at −160 mV and the calculated patch membrane area.

**Reporting summary.** Further information on research design is available in the Nature Research Reporting Summary linked to this article.

## Data availability
The data underlying the findings of this study are available from the corresponding author upon reasonable request. The source data underlying Figs. 1c–d, 2e, i, 3e–h, 4d, and 5c, e, g, Supplementary Figs. 1b, d, g, 2d, 3e, 4b, d, 5c, and 6e, and Supplementary Table 1 are provided as a Source Data file.

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

## Acknowledgements
We thank Drs Ethan M. Goldberg, Koen Vervaeke, and Iulia Glovaci for critically reading a previous version of the manuscript, Dr Thomas Klausberger for providing the physiologically relevant $PV^+$-BC AP pattern and Dr Johan F. Storm for discussions and generous equipment support. This project has received funding from the Norwegian Research Council, grant agreement number 250866, 276047 and 281252 (to H.H.) and the European Union's seventh framework program (FP7-PEOPLE-2013-COFUND), grant agreement number 609020-Scientia Fellows (to F.C.R. and H.H.).

## Author contributions
F.C.R. and H.H. performed experiments and analyzed data. H.H. wrote the first draft of the paper. Both authors jointly revised the paper.

## Competing interests
The authors declare no competing interests.
