## [Peer Review File · Nature Communications]

Reviewers' Comments:

Reviewer #1:

Remarks to the Author:

Roth and Hu investigate the function of HCN channels in the axons of GABAergic interneurons. Sub-cellular patch-clamp recordings (including outside-out patches) demonstrate that HCN channels are restricted to the axonal localization. This is a surprising finding because it is in contrast to the strong dendritic localization in hippocampal and cortical pyramidal cells. Next, the authors investigate the function of HCN channels in the axon. They find that blocking HCN channels decreases the action potential conduction velocity, which has been described before. The subsequent mechanistic investigation, however, goes beyond the current understanding in the literature. It is convincingly shown that HCN channels counter-balance the activity-dependent hyperpolarization produced by Na⁺-K⁺-ATPases.

The manuscript contains an impressive set of technically challenging direct patch-clamp recordings from axons of GABAergic interneurons allowing the authors to directly measure the interplay between HCN channels and the Na⁺-K⁺-ATPases. The manuscript is clearly written and the data analysis is solid. In summary, the manuscript represents a technically excellent tour de force with an interesting biological message, which is of interest for a broader readership. I only have the following minor comments.

In Fig. 1c and d the channel density is negative. Some data sets seem to be significantly smaller than zero. Therefore, this is probably not only noise but indicates a systematic error in the determination of the density. Also, please explain better what "by calculating chord conductance values from steady-state current amplitudes" in the methods means.

The authors sometimes refer to other publication with "we have found..." (e.g. end of 2nd paragraph in the introduction), but the list of authors is not identical. Please rephrase.

Page 15, please explain in more detail how the pipette capacitance compensation for the current-clamp recordings was performed.

Please always indicate the exact concentration of the used pharmacological blockers (e.g. ZD7288 in Fig. 3, which is used between 10 and 20 μ M according to the methods).

Reviewer #2:

Remarks to the Author:

This is an excellent paper. The authors show that in GABAergic hippocampal basket cells, HCN channels are present at high density in the axonal membrane, but are absent from the soma and dendrites. Interestingly, this is in contrast to neocortical pyramidal neurons, where HCN channels are present at highest density in dendrites. They then present a thorough and beautiful analysis of the functional role of the axonal HCN channels in counter-acting axonal hyperpolarization resulting from Na/K-ATPase pump currents during trains of action potentials, using dual recordings from soma and axon. They show that with HCN channels inhibited, the axonal membrane can hyperpolarize to near -100 mV during trains of action potentials, and that this hyperpolarization can be prevented by brief application of ouabain. The experiments are done with the highest level of technical excellence and the results are very interesting. Although the idea that HCN current can counteract post-tetanic hyperpolarization from Na/K-ATPase currents is not new, this is by far the most detailed examination of the effect by direct axonal recordings, and the demonstration that the post-tetanic hyperpolarization can result in spike failure is striking, as are the measurements of how conduction velocity is slowed with HCN blocked. The paper is very nicely written and I can find little to suggest improving.

Details:

I thought the demonstration that the membrane voltage can reach -100 mV with HCN channels inhibited was quite remarkable, and worthy of being mentioned in the Abstract. I suppose a possible caveat is that in most experiments holding currents were applied to adjust the resting potential in ZD7288 to that in control. Perhaps that is why the authors report the effect as change in membrane voltage and do not emphasize the absolute voltages. If there were experiments in which membrane voltage during trains after ZD7288 were measured with no holding current, it might be worth mentioning how negative the voltage could go.

Fig 1d- How was the surface area of the patches measured?

Point-by-point response to reviewers' comments

NCOMMS-19-38172A: An axon-specific expression of HCN channels catalyzes fast action potential signaling in GABAergic interneurons

General statement

We would like to thank the editor and both reviewers for evaluating our manuscript and providing constructive comments. We have performed experiments and analyses to address the comments raised by the reviewers and revised the manuscript text. In addition, we have made the following changes to comply with formatting requirements of Nature Communications.

- Rewriting the results in the Abstract in present tense
- Reducing the length of subheadings in the Results and Methods sections
- Updating numbering of citations of Supplementary References

Following the advice of the Editor, we have highlighted all the changes in the text in yellow.

Point-by-point reply to reviewer #1

Roth and Hu investigate the function of HCN channels in the axons of GABAergic interneurons. Sub-cellular patch-clamp recordings (including outside-out patches) demonstrate that HCN channels are restricted to the axonal localization. This is a surprising finding because it is in contrast to the strong dendritic localization in hippocampal and cortical pyramidal cells. Next, the authors investigate the function of HCN channels in the axon. They find that blocking HCN channels decreases the action potential conduction velocity, which has been described before. The subsequent mechanistic investigation, however, goes beyond the current understanding in the literature. It is convincingly shown that HCN channels counter-balance the activity-dependent hyperpolarization produced by $\text{Na}^+\text{-K}^+\text{-ATPases}$.

The manuscript contains an impressive set of technically challenging direct patch-clamp recordings from axons of GABAergic interneurons allowing the authors to directly measure the interplay between HCN channels and the $\text{Na}^+\text{-K}^+\text{-ATPases}$. The manuscript is clearly written and the data analysis is solid. In summary, the manuscript represents a technically excellent tour de force with an interesting biological message, which is of interest for a broader readership. I only have the following minor comments.

We thank the reviewer for his / her positive comments ('impressive', 'with an interesting biological message', 'of interest for a broader readership').

1) In Fig. 1c and d the channel density is negative. Some data sets seem to be significantly smaller than zero. Therefore, this is probably not only noise but indicates a systematic error in the determination of the density.

We thank the reviewer for pointing out this important detail. We have addressed the reviewer's comment with two different approaches.

First, we have performed statistical analyses to demonstrate that the somatic and dendritic h-conductance (g_h) densities in Fig. 1c and d are not significantly different from zero (somatic g_h density = -2.6 ± 1.5 pS μm^{-2} and dendritic g_h density = -1.6 ± 0.8 pS μm^{-2} , $P = 0.73$ and 0.29 , respectively, one sample sign test) despite that a subset of somatic and dendritic recordings display negative g_h density values. These results support our main conclusion that HCN channels are absent from PV⁺-BC somata and dendrites.

Second, we agree with the reviewer that the negative g_h density values in a subset of somatic and dendritic recordings may not represent biological noise. H-currents in outside-out patch experiments illustrated in Fig. 1b–d were recorded in a K⁺-rich solution containing 20 mM TEA, 5 mM 4-AP, and 1 mM BaCl₂ to block voltage-gated K⁺ currents. Because we assigned positive amplitude values to inward h-currents and negative amplitude values to outward currents in outside-out patches, the negative g_h density values may reflect a contribution from the deactivation of residual K⁺ currents in a subset of recordings. We have performed pharmacological experiments to determine whether the residual current may create a systematic error. We found that bath application of the HCN channel blocker cesium (4 mM CsCl) had no effect on currents in somatic and dendritic patches but inhibited inward currents in axonal outside-out patches. These results provide qualitative evidence to argue against that the residual current may introduce an error by masking h-currents in somatic and dendritic patches. To quantitatively estimate the inaccuracy introduced by the residual current to the measurement of g_h density, we compared the g_h density with the density of the Cs-sensitive conductance in PV⁺-BC dendrites, somata and axons (Cs-sensitive conductance density = 0.8 ± 0.8 pS μm^{-2} in dendrites, 0.3 ± 0.7 pS μm^{-2} in somata and 14.5 ± 3.7 pS μm^{-2} in axons, $n = 6$ dendritic, 6 somatic and 7 axonal outside-out patches, respectively). Because the g_h densities in Fig. 1c and d are not

significantly different from the Cs-sensitive conductance densities ($P = 0.14\text{--}0.66$, Wilcoxon rank sum test), we believe that the inaccuracy introduced by the residual current has been minimized in our recording conditions and is unlikely to create a systematic error in the results illustrated in Fig. 1c and d.

HCN channels are only relatively resistant to K^+ channel blockers^{1, 2}. We did not use higher concentrations of TEA, 4-AP and $BaCl_2$ to eliminate the residual current because this approach could potentially inhibit h-currents and cause an underestimation of h-conductance density.

We have added a description of the effect of CsCl on currents in somatic patches (page 4 of the revised manuscript). Furthermore, we have added the following sentence to the Methods section: 'To facilitate the quantification of h-current amplitudes, we assigned inward currents positive amplitude values. Conversely, negative amplitude values were assigned to outward currents.' (page 20 of the revised manuscript). In addition, we have described that the negative conductance values may reflect the contribution from residual K^+ currents (page 18 of the revised manuscript). Finally, we have added the analysis of the Cs-sensitive conductance density (page 18 of the revised manuscript).

2) Also, please explain better what "by calculating chord conductance values from steady-state current amplitudes" in the methods means.

We thank the reviewer for this constructive suggestion. We have changed the sentence in the Methods section to 'To determine h-conductance values in excised patches and whole-axon recordings, we divided the steady-state h-current amplitude by the voltage difference between the test voltage pulse and the h-current reversal potential' (page 20 of the revised manuscript).

3) The authors sometimes refer to other publication with "we have found..." (e.g. end of 2nd paragraph in the introduction), but the list of authors is not identical. Please rephrase.

We thank the reviewer for pointing out this inaccuracy. We have changed the sentence to 'Furthermore, subcellular patch-clamp recordings have revealed that the speed and reliability of AP propagation in PV^+ -BC axons is maintained during a long train of high-frequency APs' (page 3 of the revised manuscript).

4) Page 15, please explain in more detail how the pipette capacitance compensation for the current-clamp recordings was performed.

We thank the reviewer for this important comment. We have added the following sentence to the Methods section: 'Pipette capacitance compensation in current-clamp recordings was performed by adjusting the pipette capacitance neutralization function of the amplifier to the highest value without provoking oscillations inside the amplifier circuit' (page 16 of the revised manuscript).

5) Please always indicate the exact concentration of the used pharmacological blockers (e.g. ZD7288 in Fig. 3, which is used between 10 and 20 μ M according to the methods).

We thank the reviewer for this excellent comment. We have added the respective concentration of the used pharmacological blockers to the legend of every figure and to the description of experiments that are not illustrated in figures.

Point-by-point reply to reviewer #2

This is an excellent paper. The authors show that in GABAergic hippocampal basket cells, HCN channels are present at high density in the axonal membrane, but are absent from the soma and dendrites. Interestingly, this is in contrast to neocortical pyramidal neurons, where HCN channels are present at highest density in dendrites. They then present a thorough and beautiful analysis of the functional role of the axonal HCN channels in counter-acting axonal hyperpolarization resulting from Na/K-ATPase pump currents during trains of action potentials, using dual recordings from soma and axon. They show that with HCN channels inhibited, the axonal membrane can hyperpolarize to near -100 mV during trains of action potentials, and that this hyperpolarization can be prevented by brief application of ouabain. The experiments are done with the highest level of technical excellence and the results are very interesting. Although the idea that HCN current can counteract post-tetanic hyperpolarization from Na/K-ATPase currents is not new, this is by far the most detailed examination of the effect by direct axonal recordings, and the demonstration that the post-tetanic hyperpolarization can result in spike failure is striking, as are the measurements of how conduction velocity is slowed with HCN blocked. The paper is very nicely written and I can find little to suggest improving.

We thank the reviewer for his / her positive comments and finding our manuscript interesting ('excellent', 'with the highest level of technical excellence', 'interesting').

1) I thought the demonstration that the membrane voltage can reach -100 mV with HCN channels inhibited was quite remarkable, and worthy of being mentioned in the Abstract.

We thank the reviewer for this important comment. We have revised a section in the Abstract to:

‘HCN channels not only enhance AP initiation during sustained high-frequency firing but also speed up the propagation of AP trains in PV⁺-BC axons by dynamically opposing the hyperpolarization produced by Na⁺-K⁺ ATPases, which can bring axonal membrane potentials to a level near –100 mV during repetitive firing’ (page 2 of revised manuscript).

2) I suppose a possible caveat is that in most experiments holding currents were applied to adjust the resting potential in ZD7288 to that in control. Perhaps that is why the authors report the effect as change in membrane voltage and do not emphasize the absolute voltages. If there were experiments in which membrane voltage during trains after ZD7288 were measured with no holding current, it might be worth mentioning how negative the voltage could go.

We thank the reviewer for highlighting the importance to describe the effect of ZD7288 on axonal membrane potentials in absolute voltages. Following this advice, we have quantified the magnitude of the hyperpolarization during spike trains after applying ZD7288 in absolute voltages and relative peak amplitudes. With all due respect, we would also like to point out that holding currents were not adjusted to compensate for the ZD7288-induced resting membrane potential hyperpolarization in experiments illustrated in Fig. 5a–c (described in line 3–4, page 16 of the previously submitted manuscript). The most negative axonal membrane potential during the spike trains was -88.8 ± 2.3 mV after applying ZD7288 (based on the 12 soma–axon recordings in Fig. 5c).

To address the reviewer’s comment ‘If there were experiments in which membrane voltage during trains after ZD7288 were measured with no holding current, it might be worth mentioning how negative the voltage could go’, we have analyzed 5 soma–axon recordings in which no steady-state current was injected into the cell throughout the experiment. Comparison of the most negative axonal membrane potential during spike trains in these 5 experiments (-83.6 ± 2.4 mV) with that from the experiments in Fig. 5c (-88.8 ± 2.3 mV, 12 soma–axon recordings) revealed no significant difference ($P = 0.23$, Wilcoxon rank sum test). Given that the resting potential in ZD7288 was not readjusted with holding currents to match that in control during any of the experiments in Fig. 5c, we prefer using this set of recordings to describe the hyperpolarizing effect of ZD7288 during spike trains in absolute voltages as well as in relative peak amplitudes for consistency.

We have added a sentence to the Results section: 'Detailed analyses of axonal membrane potential trajectories during repetitive firing revealed that PV⁺-BC axons reached a maximum negative potential of -88.8 ± 2.3 mV (ranging between -77.7 and -101.7 mV in 12 soma–axon recordings) after inhibiting HCN channels with 20 μ M ZD7288 (Fig. 5a)' (page 7 of revised manuscript).

3) Fig 1d- How was the surface area of the patches measured?

We thank the reviewer for this important comment. We have revised the last sentence of the Methods section to provide a more detailed description of how we quantified the surface area of the patches. The original description has been replaced with the following sentences: 'The membrane area of outside-out patches was quantified with a previously established relation between patch area and pipette conductance: $A = 0.08271 \times g_P - 0.47526$, where A is patch area (μm^2) and g_P is pipette conductance (nS). Conductance density values (Fig. 1c, d) were determined from the steady-state h-current amplitude at -160 mV and the calculated patch membrane area' (page 21 of the revised manuscript).

Additional changes to the manuscript

We apologize for the incorrect description of the number of recordings in Fig. 2e and i legends (page 10–11 of the previously submitted manuscript) and of two P values in the Supplementary Information ('P = 0.01' in Supplementary Fig. 1g legend and 'P = 0.04' in Supplementary Fig. 6e legend of the previously submitted version).

We have replaced the previous text with correct values ('Data points indicate mean \pm SEM from 9–12 experiments' in Fig. 2e and 'n = 13' in Fig. 2i legends, page 11 of the revised manuscript; 'P = 0.02' in Supplementary Fig. 1g and 6e legends of the revised Supplementary Information). The correction of P values has no impact on the conclusions drawn from the Wilcoxon signed rank test.

Additional references for reviewers

1. Ludwig, A., Zong, X., Jeglitsch, M., Hofmann, F., & Biel, M. A family of hyperpolarization-activated mammalian cation channels. *Nature* **393**, 587-591 (1998).

2. Ludwig, A. et al. Two pacemaker channels from human heart with profoundly different activation kinetics. *EMBO J.* **18**, 2323-2329 (1999).

Reviewers' Comments:

Reviewer #1:

Remarks to the Author:

The authors have convincingly addressed the reviewers' comments. I fully support publications.

Reviewer #2:

Remarks to the Author:

The revised paper is mostly fine. However, with regard to the issue of absolute membrane potentials after repetitive firing in the presence of ZD7288, it seems from the authors' response and the description in the Methods ("PV + -BCs under control conditions were adjusted to a membrane potential of ~ -65 mV by injecting a holding current at the soma (range: -150 to $+200$ pA) or at the axon (range: -100 to -10 pA). " that there was always a steady negative current injection in the axon in these measurements. This then makes interpreting the absolute membrane potentials problematic – for all we know, a voltage of -100 mV with a steady current injection of -10 pA could correspond to only -80 mV if there were no current injection into the axon. This being true, I think the authors should delete the statement in the Abstract "which can bring axonal membrane potentials to a level near -100 mV during repetitive firing". Also, they should indicate in the legend for Fig 5A what was the steady current injection into the axon during this experiment, if they have that information.

Point-by-point response to reviewers' comments

NCOMMS-19-38172B: An axon-specific expression of HCN channels catalyzes fast action potential signaling in GABAergic interneurons

General statement

We would like to thank the editor and both reviewers for evaluating our manuscript and providing constructive comments. We have addressed the comments raised by reviewer #2 and revised the manuscript text accordingly. Additional changes were made to comply with the formatting requirements of Nature Communications.

Following the instructions of the editor, we are submitting a new version of the manuscript as a Word file with the use of 'Track Changes'.

Point-by-point reply to reviewer #1

The authors have convincingly addressed the reviewers' comments. I fully support publications.

We thank the reviewer for his / her positive comment.

Point-by-point reply to reviewer #2

The revised paper is mostly fine. However, with regard to the issue of absolute membrane potentials after repetitive firing in the presence of ZD7288, it seems from the authors' response and the description in the Methods ("PV⁺-BCs under control conditions were adjusted to a membrane potential of ~ -65 mV by injecting a holding current at the soma (range: -150 to $+200$ pA) or at the axon (range: -100 to -10 pA). " that there was always a steady negative current injection in the axon in these measurements. This then makes interpreting the absolute membrane potentials problematic – for all we know, a voltage of -100 mV with a steady current injection of -10 pA could correspond to only -80 mV if there were no current injection into the axon. This being true, I think the authors should delete the statement in the Abstract "which can bring axonal membrane potentials to a level near -100 mV during repetitive firing". Also, they should indicate in the legend for Fig 5A what was the steady current injection into the axon during this experiment, if they have that information.

We thank the reviewer for his / her positive comments ('The revised paper is mostly fine').

1) However, with regard to the issue of absolute membrane potentials after repetitive firing in the presence of ZD7288, it seems from the authors' response and the description in the Methods ("PV⁺-BCs under control

conditions were adjusted to a membrane potential of ~ -65 mV by injecting a holding current at the soma (range: -150 to $+200$ pA) or at the axon (range: -100 to -10 pA). “ that there was always a steady negative current injection in the axon in these measurements. This then makes interpreting the absolute membrane potentials problematic – for all we know, a voltage of -100 mV with a steady current injection of -10 pA could correspond to only -80 mV if there were no current injection into the axon. I think the authors should delete the statement in the Abstract “which can bring axonal membrane potentials to a level near -100 mV during repetitive firing”

We thank the reviewer for this important comment. Following the advice, we have removed the statement ‘which can bring axonal membrane potentials to a level near -100 mV during repetitive firing’ and revised the sentence in the Abstract to:

‘HCN channels not only enhance AP initiation during sustained high-frequency firing but also speed up the propagation of AP trains in PV⁺-BC axons by dynamically opposing the hyperpolarization produced by Na⁺-K⁺ ATPases’ (page 2 of the revised manuscript).

2) Also, they should indicate in the legend for Fig 5A what was the steady current injection into the axon during this experiment, if they have that information.

We thank the reviewer for highlighting the importance of describing the amplitude of the steady-state current injected into the axon. We have added a sentence to the legend of Figure 5a: ‘To facilitate the comparison with other recordings, we adjusted the baseline axonal membrane potential in control to -66 mV by injecting a constant hyperpolarizing current of -19.5 pA into the axon during the soma–axon recording. This holding current was not changed throughout the experiment’ (page 26–27 of the revised manuscript).